# MULTIBENCH: Multiscale Benchmarks for Multimodal Representation Learning

**Paul Pu Liang**[1], **Yiwei Lyu**[1], **Xiang Fan**[1], **Zetian Wu**[2], **Yun Cheng**[1],
**Jason Wu**[1], **Leslie Chen**[3], **Peter Wu**[1], **Michelle A. Lee**[4], **Yuke Zhu**[5],
**Ruslan Salakhutdinov**[1], **Louis-Philippe Morency**[1]
[1]CMU, [2]Johns Hopkins, [3]Northeastern, [4]Stanford, [5]UT Austin
https://cmu-multicomp-lab.github.io/multibench/

## Abstract

Learning multimodal representations involves integrating information from multiple heterogeneous sources of data. It is a challenging yet crucial area with numerous real-world applications in multimedia, affective computing, robotics, finance, human-computer interaction, and healthcare. Unfortunately, multimodal research has seen limited resources to study (1) generalization across domains and modalities, (2) complexity during training and inference, and (3) robustness to noisy and missing modalities. In order to accelerate progress towards understudied modalities and tasks while ensuring real-world robustness, we release MULTIBENCH, a systematic and unified large-scale benchmark for multimodal learning spanning 15 datasets, 10 modalities, 20 prediction tasks, and 6 research areas. MULTIBENCH provides an automated end-to-end machine learning pipeline that simplifies and standardizes data loading, experimental setup, and model evaluation. To enable holistic evaluation, MULTIBENCH offers a comprehensive methodology to assess (1) generalization, (2) time and space complexity, and (3) modality robustness. MULTIBENCH introduces impactful challenges for future research, including scalability to large-scale multimodal datasets and robustness to realistic imperfections. To accompany this benchmark, we also provide a standardized implementation of 20 core approaches in multimodal learning spanning innovations in fusion paradigms, optimization objectives, and training approaches. Simply applying methods proposed in different research areas can improve the state-of-the-art performance on 9/15 datasets. Therefore, MULTIBENCH presents a milestone in unifying disjoint efforts in multimodal machine learning research and paves the way towards a better understanding of the capabilities and limitations of multimodal models, all the while ensuring ease of use, accessibility, and reproducibility. MULTIBENCH, our standardized implementations, and leaderboards are publicly available, will be regularly updated, and welcomes inputs from the community.

## 1 Introduction

Our perception of the natural world surrounding us involves multiple sensory modalities: we see objects, hear audio signals, feel textures, smell fragrances, and taste flavors. A *modality* refers to a way in which a signal exists or is experienced. Multiple modalities then refer to a combination of multiple signals each expressed in heterogeneous manners [10]. Many real-world research problems are inherently multimodal: from the early research on audio-visual speech recognition [48] to the recent explosion of interest in language, vision, and video understanding [48] for applications such as multimedia [102, 116], affective computing [101, 127], robotics [84, 91], finance [70], dialogue [126], human-computer interaction [47, 117], and healthcare [51, 172]. The research field of multimodal machine learning (ML) brings unique challenges for both computational and theoretical research given the heterogeneity of various data sources [10]. At its core lies the learning of *multimodal representations* that capture correspondences between modalities for prediction, and has emerged as a vibrant interdisciplinary field of immense importance and with extraordinary potential.

35th Conference on Neural Information Processing Systems (NeurIPS 2021) Track on Datasets and Benchmarks.

Figure 1: MULTIBENCH contains a diverse set of 15 datasets spanning 10 modalities and testing for more than 20 prediction tasks across 6 distinct research areas, thereby enabling standardized, reliable, and reproducible large-scale benchmarking of multimodal models. To reflect real-world requirements, MULTIBENCH is designed to holistically evaluate (1) performance across domains and modalities, (2) complexity during training and inference, and (3) robustness to noisy and missing modalities.

**Limitations of current multimodal datasets:** Current multimodal research has led to impressive advances in benchmarking and modeling for specific domains such as language and vision [4, 103, 105, 132]. However, other domains, modalities, and tasks are relatively understudied. Many of these tasks are crucial for real-world intelligence such as improving accessibility to technology for diverse populations [62], accelerating healthcare diagnosis to aid doctors [78], and building reliable robots that can engage in human-AI interactions [16, 83, 137]. Furthermore, current benchmarks typically focus on performance without quantifying the potential drawbacks involved with increased time and space complexity [148], and the risk of decreased robustness from imperfect modalities [101, 123]. In real-world deployment, a balance between performance, robustness, and complexity is often required.

**MULTIBENCH:** In order to accelerate research in building general-purpose multimodal models, our main contribution is MULTIBENCH (Figure 1), a systematic and unified large-scale benchmark that brings us closer to the requirements of real-world multimodal applications. MULTIBENCH is designed to comprehensively evaluate 3 main components: generalization across domains and modalities, complexity during training and inference, and robustness to noisy and missing modalities:

1. *Generalization across domains and modalities:* MULTIBENCH contains a diverse set of 15 datasets spanning 10 modalities and testing for 20 prediction tasks across 6 distinct research areas. These research areas include important tasks understudied from a multimodal learning perspective, such as healthcare, finance, and HCI. Building upon extensive data-collection efforts by domain experts, we worked with them to adapt datasets that reflect real-world relevance, present unique challenges to multimodal learning, and enable opportunities in algorithm design and evaluation.

2. *Complexity during training and inference:* MULTIBENCH also quantifies potential drawbacks involving increased time and space complexity of multimodal learning. Together, these metrics summarize the tradeoffs of current models as a step towards efficiency in real-world settings [142].

3. *Robustness to noisy and missing modalities:* Different modalities often display different noise topologies, and real-world multimodal signals possibly suffer from missing or noisy data in at least one of the modalities [10]. MULTIBENCH provides a standardized way to assess the risk of decreased robustness from imperfect modalities through a set of modality-specific and multimodal imperfections that reflect real-world noise, thereby providing a benchmark towards safe and robust deployment.

Together, MULTIBENCH unifies efforts across separate research areas in multimodal learning to enable quick and accurate benchmarking across a wide range of datasets and metrics.

To help the community accurately compare performance and ensure reproducibility, MULTIBENCH includes an end-to-end pipeline including data preprocessing, dataset splits, multimodal algorithms, evaluation metrics, and cross-validation protocols. This includes an implementation of 20 core multimodal approaches spanning innovations in fusion paradigms, optimization objectives, and training approaches in a standard public toolkit called MULTIZOO. We perform a systematic evaluation and show that directly applying these methods can improve the state-of-the-art performance on 9 out of the 15 datasets. Therefore, MULTIBENCH presents a step towards unifying disjoint efforts in multimodal research and paves a way towards a deeper understanding of multimodal models. Most importantly, our public zoo of multimodal benchmarks and models will ensure ease of use, accessibility, and reproducibility. Finally, we outline our plans to ensure the continual availability, maintenance, and expansion of MULTIBENCH, including using it as a theme for future workshops and competitions and to support the multimodal learning courses taught around the world.

Table 1: MULTIBENCH provides a comprehensive suite of 15 multimodal datasets to benchmark current and proposed approaches in multimodal representation learning. It covers a diverse range of research areas, dataset sizes, input modalities (in the form of $\ell$: language, $i$: image, $v$: video, $a$: audio, $t$: time-series, $ta$: tabular, $f$: force sensor, $p$: proprioception sensor, $s$: set, $o$: optical flow), and prediction tasks. We provide a standardized data loader for datasets in MULTIBENCH, along with a set of state-of-the-art multimodal models.

| Research Area | Size | Dataset | Modalities | # Samples | Prediction task |
|---|---|---|---|---|---|
| Affective Computing | S | MUSTARD [24] | $\{\ell, v, a\}$ | 690 | sarcasm |
| | M | CMU-MOSI [181] | $\{\ell, v, a\}$ | 2,199 | sentiment |
| | L | UR-FUNNY [64] | $\{\ell, v, a\}$ | 16,514 | humor |
| | L | CMU-MOSEI [183] | $\{\ell, v, a\}$ | 22,777 | sentiment, emotions |
| Healthcare | L | MIMIC [78] | $\{t, ta\}$ | 36,212 | mortality, ICD-9 codes |
| Robotics | M | MUJOCO PUSH [90] | $\{i, f, p\}$ | 37,990 | object pose |
| | L | VISION&TOUCH [92] | $\{i, f, p\}$ | 147,000 | contact, robot pose |
| Finance | M | STOCKS-F&B | $\{t \times 18\}$ | 5,218 | stock price, volatility |
| | M | STOCKS-HEALTH | $\{t \times 63\}$ | 5,218 | stock price, volatility |
| | M | STOCKS-TECH | $\{t \times 100\}$ | 5,218 | stock price, volatility |
| HCI | S | ENRICO [93] | $\{i, s\}$ | 1,460 | design interface |
| Multimedia | S | KINETICS400-S [80] | $\{v, a, o\}$ | 2,624 | human action |
| | M | MM-IMDB [8] | $\{\ell, i\}$ | 25,959 | movie genre |
| | M | AV-MNIST [161] | $\{i, a\}$ | 70,000 | digit |
| | L | KINETICS400-L [80] | $\{v, a, o\}$ | 306,245 | human action |

## 2 MULTIBENCH: The MULTISCALE MULTIMODAL BENCHMARK

**Background:** We define a modality as a single particular mode in which a signal is expressed or experienced. Multiple modalities then refer to a combination of multiple heterogeneous signals [10]. The first version of MULTIBENCH focuses on benchmarking algorithms for *multimodal fusion*, where the main challenge is to join information from two or more modalities to perform a prediction (e.g., classification, regression). Classic examples for multimodal fusion include audio-visual speech recognition where visual lip motion is fused with speech signals to predict spoken words [48]. Multimodal fusion can be contrasted with multimodal translation where the goal is to generate a new and different modality [162], grounding and question answering where one modality is used to query information in another (e.g., visual question answering [4]), and unsupervised or self-supervised multimodal representation learning [109, 143]. We plan future versions of MULTIBENCH to study these important topics in multimodal research in Appendix I.

Each of the following 15 datasets in MULTIBENCH contributes a unique perspective to the various technical challenges in multimodal learning involving learning and aligning complementary information, scalability to a large number of modalities, and robustness to realistic real-world imperfections.

### 2.1 Datasets

Table 1 shows an overview of the datasets provided in MULTIBENCH. We provide a brief overview of the modalities and tasks for each of these datasets and refer the reader to Appendix C for details.

**Affective computing** studies the perception of human affective states (emotions, sentiment, and personalities) from our natural display of multimodal signals spanning language (spoken words), visual (facial expressions, gestures), and acoustic (prosody, speech tone) [124]. It has broad impacts towards building emotionally intelligent computers, human behavior analysis, and AI-assisted education. MULTIBENCH contains 4 datasets involving fusing *language*, *video*, and *audio* time-series data to predict sentiment (CMU-MOSI [181]), emotions (CMU-MOSEI [183]), humor (UR-FUNNY [64]), and sarcasm (MUSTARD [24]). Complementary information may occurs at different moments, requiring models to address the multimodal challenges of grounding and alignment.

**Healthcare:** Modern medical decision-making often involves integrating complementary information and signals from several sources such as lab tests, imaging reports, and patient-doctor conversations. Multimodal models can help doctors make sense of high-dimensional data and assist them in the diagnosis process [5]. MULTIBENCH includes the large-scale MIMIC dataset [78] which records ICU patient data including *time-series* data measured every hour and other demographic variables (e.g., age, gender, ethnicity in the form of *tabular numerical* data). These are used to predict the

disease ICD-9 code and mortality rate. MIMIC poses unique challenges in integrating time-varying and static modalities, reinforcing the need of aligning multimodal information at correct granularities.

**Robotics:** Modern robot systems are equipped with multiple sensors to aid in their decision-making. We include the large-scale MUJOCO PUSH [90] and VISION&TOUCH [92] datasets which record the manipulation of simulated and real robotic arms equipped with *visual* (RGB and depth), *force*, and *proprioception* sensors. In MUJOCO PUSH, the goal is to predict the pose of the object being pushed by the robot end-effector. In VISION&TOUCH, the goal is to predict action-conditional learning objectives that capture forward dynamics of the different modalities (contact prediction and robot end-effector pose). Robustness is important due to the risk of real-world sensor failures [89].

**Finance:** We gathered historical stock data from the internet to create our own dataset for financial time-series prediction across 3 groups of correlated stocks: STOCKS-F&B, STOCKS-HEALTH, and STOCKS-TECH. Within each group, the previous stock prices of a set of stocks are used as multimodal *time-series* inputs to predict the price and volatility of a related stock (e.g., using Apple, Google, and Microsoft data to predict future Microsoft prices). Multimodal stock prediction [136] presents scalability issues due to a large number of modalities (18/63/100 vs 2/3 in most datasets), as well as robustness challenges arising from real-world data with an inherently low signal-to-noise ratio.

**Human Computer Interaction (HCI)** studies the design of computer technology and interactive interfaces between humans and computers [43]. Many real-world problems involve multimodal inputs such as language, visual, and audio interfaces. We use the ENRICO (Enhanced Rico) dataset [40, 93] of Android app screens (consisting of an *image* as well as a *set* of apps and their locations) categorized by their design motifs and collected for data-driven design applications such as design search, user interface (UI) layout generation, UI code generation, and user interaction modeling.

**Multimedia:** A significant body of research in multimodal learning has been fueled by the large availability of multimedia data (language, image, video, and audio) on the internet. MULTIBENCH includes 3 popular large-scale multimedia datasets with varying sizes and levels of difficulty: (1) AV-MNIST [161] is assembled from *images* of handwritten digits [88] and *audio* samples of spoken digits [94], (2) MM-IMDB [8] uses movie *titles*, *metadata*, and movie *posters* to perform multi-label classification of movie genres, and (3) KINETICS [80] contains *video*, *audio*, and *optical flow* of $306, 245$ video clips annotated for $400$ human actions. To ease experimentation, we split KINETICS into small and large partitions (see Appendix C).

## 2.2 Evaluation Protocol

MULTIBENCH contains evaluation scripts for the following holistic desiderata in multimodal learning:

**Performance:** We standardize evaluation using metrics designed for each dataset, including MSE and MAE for regression to accuracy, micro & macro F1-score, and AUPRC for classification.

**Complexity:** Modern ML research unfortunately causes significant impacts to energy consumption [142], a phenomenon often exacerbated in processing high-dimensional multimodal data. As a step towards quantifying energy complexity and recommending lightweight multimodal models, MULTIBENCH records the amount of information taken in bits (i.e., data size), number of model parameters, as well as time and memory resources required during the entire training process. Real-world models may also need to be small and compact to run on mobile devices [131] so we also report inference time and memory on CPU and GPU (see Appendix D.2).

**Robustness:** Real-world multimodal data is often imperfect as a result of missing entries, noise corruption, or missing modalities entirely, which calls for robust models that can still make accurate predictions despite only having access to noisy and missing signals [101, 123]. To standardize efforts in evaluating robustness, MULTIBENCH includes the following tests: (1) *Modality-specific imperfections* are independently applied to each modality taking into account its unique noise topologies (i.e., flips and crops of images, natural misspellings in text, abbreviations in spoken audio). (2) *Multimodal imperfections* capture correlations in imperfections across modalities (e.g., missing modalities, or a chunk of time missing in multimodal time-series data). We use both qualitative measures (performance-imperfection curve) and quantitative metrics [149] that summarize (1) *relative robustness* measuring accuracy under imperfections and (2) *effective robustness* measuring the *rate* of accuracy drops after equalizing for initial accuracy on clean test data (see Appendix D.3 for details).

## 3 MULTIZOO: A Zoo of Multimodal Algorithms

To complement MULTIBENCH, we release a comprehensive toolkit, MULTIZOO, as starter code for multimodal algorithms which implements 20 methods spanning different methodological innova-

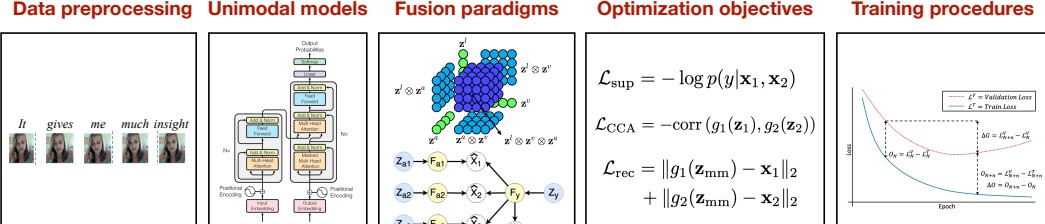

Figure 2: MULTIZOO provides a standardized implementation of a suite of multimodal methods in a modular fashion to enable accessibility for new researchers, compositionality of approaches, and reproducibility of results.

tions in (1) data preprocessing, (2) fusion paradigms, (3) optimization objectives, and (4) training procedures (see Figure 2). To introduce these algorithms, we use the simple setting with 2 modalities for notational convenience but refer the reader to Appendix E for detailed descriptions and implementations. We use $\mathbf{x}_1, \mathbf{x}_2$ for input modalities, $\mathbf{z}_1, \mathbf{z}_2$ for unimodal representations, $\mathbf{z}_{\mathrm{mm}}$ for the multimodal representation, and $\hat{y}$ for the predicted label.

## 3.1 Data Preprocessing

**Temporal alignment** [26] has been shown to help tackle the multimodal alignment problem for time-series data. This approach assumes a temporal granularity of the modalities (e.g., at the level of words for text) and aligns information from the remaining modalities to the same granularity. We call this approach WORDALIGN [26] for temporal data where text is one of the modalities.

## 3.2 Fusion Paradigms

**Early and late fusion:** Early fusion performs concatenation of input data before using a model (i.e., $\mathbf{z}_{\mathrm{mm}} = [\mathbf{x}_1, \mathbf{x}_2]$) while late fusion applies suitable unimodal models to each modality to obtain their feature representations, concatenates these features, and defines a classifier to the label (i.e., $\mathbf{z}_{\mathrm{mm}} = [\mathbf{z}_1, \mathbf{z}_2]$) [10]. MULTIZOO includes their implementations denoted as EF and LF respectively.

**Tensors** are specifically designed to tackle the multimodal complementarity challenge by explicitly capturing higher-order interactions across modalities [179]. Given unimodal representations $\mathbf{z}_1, \mathbf{z}_2$, tensors are defined as $\mathbf{z}_{\mathrm{mm}} = \begin{bmatrix} \mathbf{z}_1 \\ 1 \end{bmatrix} \otimes \begin{bmatrix} \mathbf{z}_2 \\ 1 \end{bmatrix}$ where $\otimes$ denotes an outer product. However, computing tensor products is expensive since their dimension scales exponentially with the number of modalities so several efficient approximations have been proposed [71, 101, 106]. MULTIZOO includes Tensor Fusion (TF) [179] as well as the approximate Low-rank Tensor Fusion (LRTF) [106].

**Multiplicative Interactions (MI)** generalize tensor products to include learnable parameters that capture multimodal interactions [77]. In its most general form, MI defines a bilinear product $\mathbf{z}_{\mathrm{mm}} = \mathbf{z}_1 \mathbb{W} \mathbf{z}_2 + \mathbf{z}_1^\top \mathbf{U} + \mathbf{V} \mathbf{z}_2 + \mathbf{b}$ where $\mathbb{W}, \mathbf{U}, \mathbf{Z}$, and $\mathbf{b}$ are trainable parameters. By appropriately constraining the rank and structure of these parameters, MI recovers HyperNetworks [61] (unconstrained parameters resulting in a matrix output), Feature-wise linear modulation (FiLM) [120, 188] (diagonal parameters resulting in vector output), and Sigmoid units [37] (scalar parameters resulting in scalar output). MULTIZOO includes all 3 as MI-MATRIX, MI-VECTOR, and MI-SCALAR respectively.

**Multimodal gated units** learn representations that dynamically change for every input [25, 167, 171]. Its general form can be written as $\mathbf{z}_{\mathrm{mm}} = \mathbf{z}_1 \odot h(\mathbf{z}_2)$, where $h$ represents a function with sigmoid activation and $\odot$ denotes element-wise product. $h(\mathbf{z}_2)$ is commonly referred to as "attention weights" learned from $\mathbf{z}_2$ to attend on $\mathbf{z}_1$. Attention is conceptually similar to MI-VECTOR but recent work has explored more expressive forms of $h$ such as using a Query-Key-Value mechanism [167] or fully-connected layers [25]. We implement the Query-Key-Value mechanism as NL GATE [167].

**Temporal attention models** tackle the challenge of multimodal alignment and complementarity. Transformer models [158] are useful for temporal data by automatically aligning and capturing complementary features at different time-steps [154, 174]. We include the Multimodal Transformer (MULT) [154] which applied a Crossmodal Transformer block using $\mathbf{z}_1$ to attend to $\mathbf{z}_2$ (and vice-versa) to obtain a multimodal representation $\mathbf{z}_{\mathrm{mm}} = [\mathbf{z}_{1 \to 2}, \mathbf{z}_{2 \to 1}] = [\mathrm{CM}(\mathbf{z}_1, \mathbf{z}_2), \mathrm{CM}(\mathbf{z}_2, \mathbf{z}_1)]$.

**Architecture search:** Instead of hand-designing architectures, several approaches define a set of atomic operations (e.g., linear transformation, activation, attention, etc.) and use architecture search to learn the best order of these operations for a given task [122, 173], which we call MFAS.

**Algorithm 1** PyTorch code integrating MULTIBENCH datasets and MULTIZOO models.

```
from datasets.get_data import get_dataloader
from unimodals.common_models import ResNet, Transformer
from fusions.common_fusions import MultInteractions
from training_structures.gradient_blend import train, test

# loading Multimodal IMDB dataset
traindata, validdata, testdata = get_dataloader('multimodal_imdb')
out_channels = 3
# defining ResNet and Transformer unimodal encoders
encoders = [ResNet(in_channels=1, out_channels, layers=5),
            Transformer(in_channels=1, out_channels, layers=3)]
# defining a Multiplicative Interactions fusion layer
fusion = MultInteractions([out_channels*8, out_channels*32], out_channels*32, 'matrix')
classifier = MLP(out_channels*32, 100, labels=23)
# training using Gradient Blend algorithm
model = train(encoders, fusion, classifier, traindata, validdata,
              epochs=100, optimtype=torch.optim.SGD, lr=0.01, weight_decay=0.0001)
# testing
performance, complexity, robustness = test(model, testdata)
```

## 3.3 Optimization Objectives

In addition to the standard supervised losses (e.g., cross entropy for classification, MSE/MAE for regression), several proposed methods have proposed new objective functions based on:

**Prediction-level alignment** objectives tackle the challenge of alignment by capturing a representations where semantically similar concepts from different modalities are close together [9, 33, 91, 151]. Alignment objectives have been applied at both prediction and feature levels. In the former, we implement Canonical Correlation Analysis (CCA) [7, 145, 166], which maximizes correlation by adding a loss term $\mathcal{L}_{\mathrm{CCA}} = -\mathrm{corr}(g_1(\mathbf{z}_1), g_2(\mathbf{z}_2))$ where $g_1, g_2$ are auxiliary classifiers mapping each unimodal representation to the label.

**Feature-level alignment:** In the latter, contrastive learning has emerged as a popular approach to bring similar concepts close in feature space and different concepts far away [33, 91, 151]. We include REFNET [135] which uses a self-supervised contrastive loss between unimodal representations $\mathbf{z}_1, \mathbf{z}_2$ and the multimodal representation $\mathbf{z}_{\mathrm{mm}}$, i.e., $\mathcal{L}_{\mathrm{contrast}} = 1 - \cos(\mathbf{z}_{\mathrm{mm}}, g_1(\mathbf{z}_1)) + 1 - \cos(\mathbf{z}_{\mathrm{mm}}, g_2(\mathbf{z}_2))$ where $g_1, g_2$ is a layer mapping each modality's representation into the joint multimodal space.

**Reconstruction objectives** based on generative-discriminative models (e.g., VAEs) aim to reconstruct the input (or some part of the input) [91, 155]. These have been shown to better preserve task-relevant information learned in the representation, especially in settings with sparse supervised signals such as robotics [91] and long videos [155]. We include the Multimodal Factorized Model (MFM) [155] that learns a representation $\mathbf{z}_{\mathrm{mm}}$ that can reconstruct input data $\mathbf{x}_1, \mathbf{x}_2$ while also predicting the label, i.e., adding an objective $\mathcal{L}_{\mathrm{rec}} = \|g_1(\mathbf{z}_{\mathrm{mm}}) - \mathbf{x}_1\|_2 + \|g_2(\mathbf{z}_{\mathrm{mm}}) - \mathbf{x}_2\|_2$ where $g_1, g_2$ are auxiliary decoders mapping $\mathbf{z}_{\mathrm{mm}}$ to each raw input modality. MFM can be paired with any multimodal model from section 3.2 (e.g., learning $\mathbf{z}_{\mathrm{mm}}$ via tensors and adding a term to reconstruct input data).

**Improving robustness:** These approaches modify the objective function to account for robustness to noisy [101] or missing [89, 111, 123] modalities. MULTIZOO includes MCTN [123] which uses cycle-consistent translation to predict the noisy/missing modality from present ones (i.e., a path $\mathbf{x}_1 \to \mathbf{z}_{\mathrm{mm}} \to \hat{\mathbf{x}}_2 \to \mathbf{z}_{\mathrm{mm}} \to \hat{\mathbf{x}}_1$, with additional reconstruction losses $\mathcal{L}_{\mathrm{rec}} = \|\mathbf{x}_1 - \hat{\mathbf{x}}_1\|_2 + \|\mathbf{x}_2 - \hat{\mathbf{x}}_2\|_2$). While MCTN is trained with multimodal data, it only takes in one modality $\mathbf{x}_1$ at test-time which makes it robust to the remaining modalities.

## 3.4 Training Procedures

**Improving generalization:** Recent work has found that directly training a multimodal model is sub-optimal since different modalities overfit and generalize at different rates. MULTIZOO includes Gradient Blending (GRADBLEND), that computes generalization statistics for each modality to determine their weights during fusion [167], and Regularization by Maximizing Functional Entropies (RMFE), which uses functional entropy to balance the contribution of each modality to the result [53].

## 3.5 Putting Everything Together

In Algorithm 1, we show a sample code snippet in Python that loads a dataset from MULTIBENCH (section C.2), defines the unimodal and multimodal architectures, optimization objective, and training procedures (section 3), before running the evaluation protocol (section 2.2). Our MULTIZOO toolkit is easy to use and trains entire multimodal models in less than 10 lines of code. By standardizing the implementation of each module and disentangling the individual effects of models, optimizations, and training, MULTIZOO ensures both accessibility and reproducibility of its algorithms.

Table 2: **Standardizing methods and datasets** enables quick application of methods from different research areas which achieves stronger performance on 9/15 datasets in MULTIBENCH, especially in healthcare, HCI, robotics, and finance. *In-domain* refers to the best performance across methods previously proposed on that dataset and *out-domain* shows best performance across remaining methods. ↑ indicates metrics where higher is better (Acc, AUPRC), ↓ indicates lower is better (MSE).

| Dataset | MUSTARD ↑ | CMU-MOSI ↑ | UR-FUNNY ↑ | CMU-MOSEI ↑ | MIMIC ↑ |
|---|---|---|---|---|---|
| Unimodal | 68.6 ± 0.4 | 74.2 ± 0.5 | 58.3 ± 0.2 | 78.8 ± 1.5 | 76.7 ± 0.3 |
| In-domain | 66.3 ± 0.3 | **83.0 ± 0.1** | 62.9 ± 0.2 | **82.1 ± 0.5** | 77.9 ± 0.3 |
| Out-domain | **71.8 ± 0.3** | 75.5 ± 0.5 | **66.7 ± 0.3** | 78.1 ± 0.3 | **78.2 ± 0.2** |
| Improvement | 4.7% | - | 6.0% | - | 0.4% |

| Dataset | MUJOCO PUSH ↓ | V&T EE ↓ | STOCKS-F&B ↓ | STOCKS-HEALTH ↓ | STOCKS-TECH ↓ |
|---|---|---|---|---|---|
| Unimodal | 0.334 ± 0.034 | 0.202 ± 0.022 | 1.856 ± 0.093 | 0.541 ± 0.010 | 0.125 ± 0.004 |
| In-domain | **0.290 ± 0.018** | 0.258 ± 0.011 | 1.856 ± 0.093 | 0.541 ± 0.010 | 0.125 ± 0.004 |
| Out-domain | 0.402 ± 0.026 | **0.185 ± 0.011** | **1.820 ± 0.138** | **0.526 ± 0.017** | **0.120 ± 0.008** |
| Improvement | - | 8.4% | 1.9% | 2.8% | 4.0% |

| Dataset | ENRICO ↑ | MM-IMDB ↑ | AV-MNIST ↑ | KINETICS-S ↑ | KINETICS-L ↑ |
|---|---|---|---|---|---|
| Unimodal | 47.0 ± 1.6 | 45.6 ± 4.5 | 65.1 ± 0.2 | **56.5** | 72.6 |
| In-domain | 47.0 ± 1.6 | 49.8 ± 1.7 | **72.8 ± 0.2** | 56.1 | **74.7** |
| Out-domain | **51.0 ± 1.4** | **50.2 ± 0.9** | 72.3 ± 0.2 | 23.7 | 71.7 |
| Improvement | 8.5% | 0.8% | - | - | - |

# 4 Experiments and Discussion

**Setup:** Using MULTIBENCH, we load each of the datasets and test the multimodal approaches in MULTIZOO. We only vary the contributed method of interest and keep all other possibly confounding factors constant (i.e., using the exact same training loop when testing a new multimodal fusion paradigm), a practice unfortunately not consistent in previous work. Our code is available at https://github.com/pliang279/MultiBench. Please refer to Appendix G for experimental details. MULTIBENCH allows for careful analysis of multimodal models and we summarize the main take-away messages below (see Appendix H for full results and analysis).

**Benefits of standardization:** From Table 2, simply applying methods proposed *outside* of the same research area can improve the state-of-the-art performance on 9 of the 15 MULTIBENCH datasets, especially for relatively understudied domains and modalities (i.e., healthcare, finance, HCI).

**Generalization across domains and modalities:** MULTIBENCH offers an opportunity to analyze algorithmic developments across a large suite of modalities, domains, and tasks. We summarize the following observations regarding performance across datasets and tasks (see details in Appendix H.7):

1. Many multimodal methods show their strongest performance on in-domain datasets and do not generalize across domains and modalities. For example, MFAS [122] works well on domains it was designed for (AV-MNIST and MM-IMDB in multimedia) but does not generalize to other domains such as healthcare (MIMIC). Similarly, MULT [154] performs extremely well on the affect recognition datasets it was designed for but struggles on other multimodal time-series data in the finance and robotics domains. Finally, GRADBLEND [167], an approach specifically designed to improve generalization in multimodal learning and tested on video and audio datasets (e.g., Kinetics), does not perform well on other datasets. In general, we observe high variance in the performance of multimodal methods across datasets in MULTIBENCH. Therefore, there still does not exist a one-size-fits-all model, especially for understudied modalities and tasks.

2. There are methods that are surprisingly generalizable across datasets. These are typically general modality-agnostic methods such as LF. While simple, it is a strong method that balances simplicity, performance, and low complexity. However, it does not achieve the best performance on any dataset.

3. Several methods such as MFAS and CCA are designed for only 2 modalities (usually image and text), and TF and MI do not scale efficiently beyond 2/3 modalities. We encourage the community to generalize these approaches across datasets and modalities on MULTIBENCH.

**Tradeoffs between modalities:** How far can we go with unimodal methods? Surprisingly far! From Table 2, we observe that decent performance can be obtained with the best performing modality. Further improvement via multimodal models may come at the expense of around 2−3× the parameters.

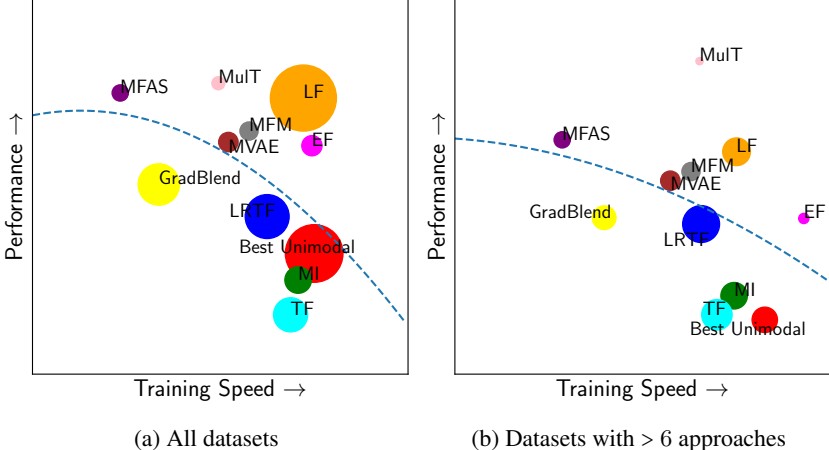

(a) All datasets                    (b) Datasets with > 6 approaches

Figure 3: **Tradeoff between performance and complexity**. Size of circles shows variance in performance across (a) all datasets and (b) datasets on which we tested > 6 approaches. We plot a dotted blue line of best quadratic fit to show the Pareto frontier. These strong tradeoffs should encourage future work in lightweight multimodal models that generalize across datasets, as well as in adapting several possibly well-performing methods (such as MFAS or MULT) to new datasets and domains.

**Tradeoffs between performance and complexity:** In Figure 3(a), we summarize the performance of all methods in terms of performance and complexity. We find a strong tradeoff between these two desiderata: simple fusion techniques (e.g., LF) are actually appealing choices which score high on both metrics, especially when compared to complex (but slightly better performing) methods such as architecture search (MFAS) or Multimodal Transformers (MULT). While LF is the easiest to adapt to new datasets and domains, we encountered difficulties in adapting several possibly well-performing methods (such as MFAS or MULT) to new datasets and domains. Therefore, while their average performance is only slightly better than LF on all datasets (see Figure 3(a)), they perform much better on well-studied datasets (see Figure 3(b)). We hope that the release of MULTIBENCH will greatly accelerate research in adapting complex methods on new datasets (see full results in Appendix H.8).

**Tradeoffs between performance and robustness:** In Figure 4, we plot a similar tradeoff plot between accuracy and (relative & effective) robustness. As a reminder, relative robustness directly measures accuracy under imperfections while effective robustness measures the rate at which accuracy drops after equalizing for initial accuracy on clean test data (see Appendix D.3 for details). We observe a positive correlation between performance and relative robustness (see Figure 4(a)), implying that models starting off with higher accuracy tend to stay above other models on the performance-imperfection curve. However, we observe a negative best fit between performance and effective robustness (see Figure 4(b)) because several well-performing methods such as MULT, CCA, and MVAE tend to *drop off faster* after equalizing for initial accuracy on clean test data. Furthermore, very few models currently achieve both positive relative and effective robustness, which is a crucial area for future multimodal research (see full results in Appendix H.9).

## 5  Related Work

We review related work on standardizing datasets and methods in multimodal learning.

**Comparisons with related benchmarks:** To the best of our knowledge, MULTIBENCH is the first multimodal benchmark with such a large number of datasets, modalities, and tasks. Most previous multimodal benchmarks have focused on a single research area such as within affective computing [56], human multimodal language [177], language and vision-based question answering [50, 138], text classification with external multimodal information [60], and multimodal learning for education [65]. MULTIBENCH is specifically designed to go beyond the commonly studied language, vision, and audio modalities to encourage the research community to explore relatively understudied modalities (e.g., tabular data, time-series, sensors, graph and set data) and build general multimodal methods that can handle a diverse set of modalities.

Our work is also inspired by recent progress in better evaluation benchmarks for a suite of important tasks in ML such as language representation learning [163, 164], long-range sequence modeling [150], multilingual representation learning [72], graph representation learning [74], and robustness to distribution shift [85]. These well-crafted benchmarks have accelerated progress in new algorithms, evaluation, and analysis in their respective research areas.

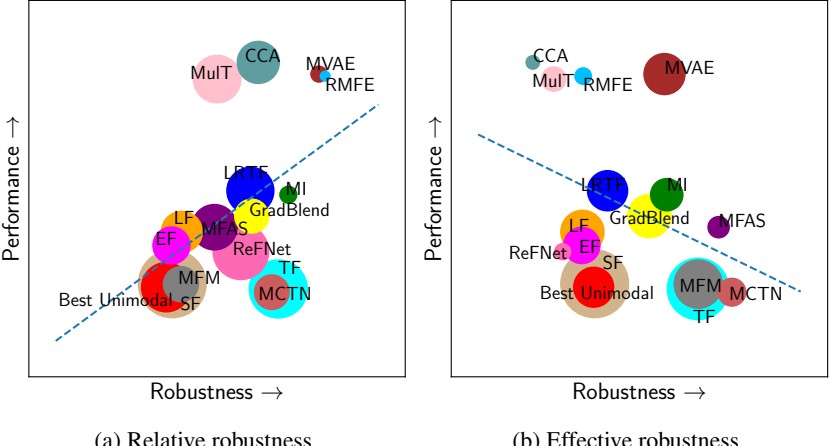

(a) Relative robustness                         (b) Effective robustness

Figure 4: **Tradeoff between performance and robustness**. Size of circles shows variance in robustness across datasets. We show the line of best linear fit in dotted blue. While better performing methods show better *relative* robustness (a), some suffer in *effective* robustness since performance *drops off faster* (b). Few models currently achieve both relative and effective robustness, which suggests directions for future research.

**Standardizing multimodal learning:** There have also been several attempts to build a single model that works well on a suite of multimodal tasks [95, 109, 143]. However, these are limited to the language and vision space, and multimodal training is highly tailored for text and images. Transformer architectures have emerged as a popular choice due to their suitability for both language and image data [27, 73] and a recent public toolkit was released for incorporating multimodal data on top of text-based Transformers for prediction tasks [60]. By going beyond Transformers and text data, MULTIBENCH opens the door to important research questions involving a much more diverse set of modalities and tasks while holistically evaluating performance, complexity, and robustness.

**Analysis of multimodal representations:** Recent work has begun to carefully analyze and challenge long-standing assumptions in multimodal learning. They have shown that certain models do not actually learn cross-modal interactions but rather rely on ensembles of unimodal statistics [68] and that certain datasets and models are biased to the most dominant modality [22, 59], sometimes ignoring others completely [3]. These observations are currently only conducted on specific datasets and models without testing their generalization to others, a shortcoming we hope to solve using MULTIBENCH which enables scalable analysis over modalities, tasks, and models.

## 6   Conclusion

**Limitations:** While MULTIBENCH can help to accelerate research in multimodal ML, we are aware of the following possible limitations (see detailed future directions in Appendix I):

1. *Tradeoffs between generality and specificity:* While it is desirable to build models that work across modalities and tasks, there is undoubtedly merit in building modality and task-specific models that can often utilize domain knowledge to improve performance and interpretability (e.g., see neuro-symbolic VQA [159], or syntax models for the language modality [31]). MULTIBENCH is not at odds with research in this direction: in fact, by easing access to data, models, and evaluation, we hope that MULTIBENCH will challenge researchers to design interpretable models leveraging domain knowledge for many multimodal tasks. It remains an open question to define "interpretability" for other modalities beyond image and text, a question we hope MULTIBENCH will drive research in.

2. *Scale of datasets, models, and metrics:* We plan for MULTIBENCH to be a continuously-growing community effort with regular maintenance and expansion. While MULTIBENCH currently does not include several important research areas outside of multimodal fusion (e.g., question answering [4, 63], retrieval [187], grounding [32], and reinforcement learning [110]), and is also limited by the models and metrics it supports, we outline our plan to expand in these directions in Appendix I.

**Projected expansions of MULTIBENCH:** In this subsection, we describe concrete ongoing and future work towards expanding MULTIBENCH (see details in Appendix I).

1. *Other multimodal research problems:* We are genuinely committed to building a community around these resources and continue improving it over time. While we chose to focus on multimodal fusion by design for this first version to have a more coherent way to standardize and evaluate methods across datasets, we acknowledge the breadth of multimodal learning and are looking forward to expanding

it in other directions in collaboration with domain experts. We have already included 2 datasets in captioning (and more generally for non-language outputs, retrieval): (1) Yummly-28K of paired videos and text descriptions of food recipes [114], and (2) Clotho dataset for audio-captioning [45] as well as a language-guided RL environment Read to Fight Monsters (RTFM) [188] and are also working towards more datasets in QA, retrieval, and multimodal RL.

To help in scalable expansion, we plan for an open call to the community for suggestions and feedback about domains, datasets, and metrics. As a step in this direction, we have concrete plans to use MULTIBENCH as a theme for future workshops and competitions (building on top of the multimodal workshops we have been organizing at NAACL 2021, ACL 2020, and ACL 2019, and in multimodal learning courses (starting with the course taught annually at CMU). Since MULTIBENCH is public and will be regularly maintained, the existing benchmark, code, evaluation, and experimental protocols can greatly accelerate any dataset and modeling innovations added in the future. In our public GitHub, we have included a section on contributing through task proposals or additions of datasets and algorithms. The authors will regularly monitor new proposals through this channel.

2. *New evaluation metrics:* We also plan to include evaluation for distribution shift, uncertainty estimation, tests for fairness and social biases, as well as labels/metrics for interpretable multimodal learning. In the latter, we plan to include the EMAP score [68] as an interpretability metric assessing whether cross-modal interactions improve performance.

3. *Multimodal transfer learning and co-learning:* Can training data in one dataset help learning on other datasets? MULTIBENCH enables easy experimentation of such research questions: our initial experiments on transfer learning found that pre-training on larger datasets in the same domain can improve performance on smaller datasets when fine-tuned on a smaller dataset: performance on the smaller CMU-MOSI dataset improved from 75.2 to 75.8 using the same late fusion model with transfer learning from the larger UR-FUNNY and CMU-MOSEI datasets. Furthermore, recent work has shown that multimodal training can help improve unimodal performance as well [140, 170, 180]. While previous experiments were on a small scale and limited to a single domain, we plan to expand significantly on this phenomenon (multimodal co-learning) in future versions of MULTIBENCH.

4. *Multitask learning across modalities:* Multitask learning across multimodal tasks with a shared set of input modalities is a promising direction that can enable statistical strength sharing across datasets and efficiency in training a single model. Using MULTIBENCH, we also ran an extra experiment on multi-dataset multitask learning. We used the 4 datasets in the affective computing domain and trained a single model across all 4 of them with adjustable input embedding layers if the input features were different and separate classification heads for each dataset's task. We found promising initial results with performance on the largest CMU-MOSEI dataset improving from 79.2 to 80.9 for a late fusion model and from 82.1 to 82.9 using a multimodal transformer model, although performance on the smaller CMU-MOSI dataset decreased from 75.2 to 70.8. We believe that these potential future studies in co-learning, transfer learning, and multi-task learning are strengths of MULTIBENCH since it shows the potential of interesting experiments and usage.

**In conclusion**, we present MULTIBENCH, a large-scale benchmark unifying previously disjoint efforts in multimodal research with a focus on ease of use, accessibility, and reproducibility, thereby paving the way towards a deeper understanding of multimodal models. Through its unprecedented range of research areas, datasets, modalities, tasks, and evaluation metrics, MULTIBENCH highlights several future directions in building more generalizable, lightweight, and robust multimodal models.

## Acknowledgements

This material is based upon work partially supported by the National Science Foundation (Awards #1722822 and #1750439), National Institutes of Health (Awards #R01MH125740, #R01MH096951, #U01MH116925, and #U01MH116923), BMW of North America, and SquirrelAI. PPL is supported by a Facebook PhD Fellowship and a Center for Machine Learning and Health Fellowship. RS is supported in part by NSF IIS1763562 and ONR Grant N000141812861. Any opinions, findings, conclusions, or recommendations expressed in this material are those of the author(s) and do not necessarily reflect the views of the National Science Foundation, National Institutes of Health, Facebook, CMLH, Office of Naval Research, BMW of North America, and SquirrelAI, and no official endorsement should be inferred. We are extremely grateful to Amir Zadeh, Chaitanya Ahuja, Volkan Cirik, Murtaza Dalal, Benjamin Eysenbach, Tiffany Min, and Devendra Chaplot for helpful discussions and feedback, as well as Ziyin Liu and Chengfeng Mao for providing tips on working with financial time-series data. Finally, we would also like to acknowledge NVIDIA's GPU support.

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
