# OpenReview forum: "MultiBench: Multiscale Benchmarks for Multimodal Representation Learning"
_NeurIPS.cc/2021/Track/Datasets_and_Benchmarks/Round1 — NeurIPS 2021 Datasets and Benchmarks Track (Round 1)_

### Official Review · Reviewer_xwdp · 2021-06-30
**Good initial attempt to introduce a multimodal benchmark dataset for representation learning, though the current version mostly focuses on evaluating fusion strategies and the selection process for the included datasets is not very transparent.**

**Rating:** 7
**Confidence:** 4

**Strengths:**

+ The paper comes with a standard toolkit enabling an easy to use and easy to extend interface for end-to-end training training of multimodal models.
+ The authors provide a model zoo consisting of implementations of 20 algorithms with various differences, and a leaderboard (not active at the time of this review).
+ The datasets show great diversity in terms of application areas, modality types as well as prediction tasks.
+ The experimental evaluation highlights the benefit of having a modular framework to test the effectiveness of certain components (e.g. fusion strategy) on a variety of datasets.
+ The paper reports state-of-the-art results on 9/15 dataset by exploring the performances of methods/components on domains/datasets not tested on before.

**Weaknesses:**

- The paper mostly focuses on evaluating different fusion strategies and algorithms. See my additional comments to the authors.
- The selection criteria of the research areas and the datasets are not explained in a transparent manner.

**Additional Feedback:**

Overall, I think the paper is in the right direction but needs a revision considering the points I raised above.

First and foremost, the decision process for including the selected 15 datasets is not clearly explained. I agree with the authors that they pose certain challenges. However, I think a better way could be to announce an open call to the community for task and dataset proposals, similar to what’s been done for the SuperGLUE benchmark (Wang et al., 2019) and the recently proposed GEM benchmark (Gehrmann et al., 2021).
(The authors carefully addressed my concerns and added a section on their plans to extend their benchmark)

As I mentioned before, the current version of the benchmark solely concentrates on benchmarking various different fusion strategies. As a person interested in multimodal research, I would like to see experiments on evaluating the performances of the methods under a multi-task learning setting. As the datasets include overlapping modalities, this could be a really interesting research direction.
(The authors added new experiments to respond my suggestion)

**Clarity:**

The paper is written well. That said, due to the comprehensiveness of the proposed benchmark, most of the technical details regarding the datasets and the models are given in the supplementary material, which decreases the readibility of the paper and there exists some repetitions.

**Correctness:**

The authors make use of already existing datasets to construct the proposed benchmark. In a nutshell, it considers a modular design, which allows for evaluating different combinations of preprocessing steps, fusion strategies and optimization schemes at ease. Moreover, it includes standard evaluation metrics per each prediction task. In those regards, the claims made and the evaluation setup considered in the submission sound correct.

**Documentation:**

The proposed benchmark is well documented. The model zoo and the scripts provided allow for reproducing the results given in the submission. Moreover, the ways to include new datasets and algorithms to the current benchmark are clearly defined.

**Ethics:**

The proposed benchmark is based on a collection of multimodal datasets existing in the literature and capturing different areas of application including healthcare. The authors explicitly stated that the collection of these datasets takes into account the ethical review processes and individuals consent. Hence, there are no any other ethical concerns that need further discussion.

**Relation To Prior Work:**

I am not aware of any such multimodal benchmark of this scale and variety.

**Summary And Contributions:**

The paper presents a new benchmark for multimodal representation learning. It is composed of 15 different datasets that exist in the literature, covering 10 distinct modalities (language, image, video, audio, etc.) and different prediction tasks. Each dataset in the benchmark is of different size and captures different application areas like affective computing, healthcare, robotics, finance, HCI and multimedia processing, which both require integration of multiple modalities. The authors construct a zoo of multimodal algorithms for data preprocessing, fusion paradigms and optimization objectives, and evaluate their performance, complexity and robustness on the suggested benchmark. The experimental analysis reveals important observations and sheds light on possible future directions for multimodal models.

---

> ### Author Response · Authors · 2021-07-11
> **Thank you for your feedback, we have revised our paper accordingly**
>
> Thank you for your valuable feedback and insightful comments! We respond to some concerns below:
>
> [readability] We apologize for the long appendix. We wanted to be as thorough as possible in our benchmark description and experimental setup+analysis. We are currently planning an arXiv version with a longer main paper with technical details regarding datasets and models, while leaving additional details in the appendix.
>
> [open call] Thank you for your suggestion. This is aligned with our future plans. We agree that an open call to the community for task and dataset proposals is a scalable method in the expansion of MultiBench. Getting a first version of the MultiBench dataset will allow people to understand what can be done (in this case for the multimodal fusion sub-problem) and enable a discussion on which directions to expand the dataset. We have concrete plans to use MultiBench as a theme for future workshops and competitions (*) and offer an open-call for suggestions and feedback about domains, datasets and metrics, and have added details regarding this in Appendix I.4.
>
> (*) building from our prior experience organizing multimodal workshops at NAACL 2021: http://multicomp.cs.cmu.edu/naacl2021multimodalworkshop, ACL 2020: http://multicomp.cs.cmu.edu/acl2020multimodalworkshop, and ACL 2019: http://multicomp.cs.cmu.edu/acl2018multimodalchallenge), and in multimodal learning courses (starting with the course taught annually at CMU: https://cmu-multicomp-lab.github.io/mmml-course/fall2020).
>
> Since MultiBench is publicly released and will be regularly maintained, the existing starting benchmark, code, evaluation, and experimental protocols can greatly accelerate the addition of new datasets and models in the future. In the public GitHub (https://github.com/pliang279/MultiBench), we have also included a section on contributing to MultiBench through either task proposals or additions of datasets and algorithms. The authors will regularly monitor new proposals through this channel.
>
> [only fusion] We are genuinely committed to building a community around these resources and continue improving it over time. We chose to focus on multimodal fusion by design for this first version of MultiBench to have a more coherent way to standardize and evaluate methods across datasets. We acknowledge the breath of multimodal learning and are looking forward to expanding it in other directions. We already identified 3 candidates for this expansion: multimodal grounding (e.g., image, video, and table-based question answering), cross-modal translation (e.g., image captioning and retrieval) and reinforcement learning (including language-vision navigation and robotics). We have already included 2 datasets in captioning (and more generally for non-language outputs, retrieval): (1) Yummly-28K of paired videos and text descriptions of food recipes [1], and (2) Clotho dataset for audio-captioning [2] in our latest GitHub commits (https://github.com/pliang279/MultiBench/commits/main). We have also included a language-guided RL environment Read to Fight Monsters (RTFM) [3] in the recent commits too (see details on added text in page 10, under 'Other multimodal research problems'). For this purpose, we plan an open call so that we can get feedback from the community (see Appendix I.4).
>
> [multitask] We agree that multitask learning across multimodal tasks with a shared set of input modalities is a very interesting research problem. Therefore, we ran an extra experiment to test for multi-dataset multitask learning. To do so, we used the 4 datasets in the affective computing domain and trained a single model across all 4 of them with adjustable input embedding layers if the input features were different and separate classification heads for each dataset’s task. We found promising initial results with performance on the largest CMU-MOSEI dataset improving from 79.2 to 80.9 for a late fusion model and from 82.1 to 82.9 using a multimodal transformer model, although performance on the smaller CMU-MOSI dataset decreased from 75.2 to 70.8. We believe that these potential future studies in transfer learning and multi-task learning are in fact strengths of MultiBench since it enables easy experimentation of such research questions. (see details on added text in page 10, under 'Multitask learning across modalities').
>
> In fact, quite a few of our current experiments do use multitask learning within a single dataset - for example in MIMIC healthcare dataset we train a single model to predict the 3 tasks (mortality, ICD-9 10-19, and ICD-9 70-79) at the same time, and for CMU-MOSEI affective computing dataset we train a single model to predict all 9 discrete emotions (angry, excited, fear, sad, surprised, frustrated, happy, disappointed, and neutral) at the same time. We found that multitask training improved upon training for each task separately.

---

### Official Review · Reviewer_jKQc · 2021-07-02
**A meaningful benchmark for multimodal learning**

**Rating:** 6
**Confidence:** 4
**Correctness:** It looks correct.
**Clarity:** Well written.

**Strengths:**

1. The benchmark is well motivated. The writing is clear and easy to understand.
2. The evaluation of the benchmark is sufficient. It analyzes the performance of existing well-known techniques in feature interaction and optimization. The reviewer does get some new insights about these well-known techniques.


**Weaknesses:**

The reviewer does have some concerns about the paper.
1. Although the benchmark contains about 10 modalities. The benchmark mainly focuses on simple classification tasks. The reviewer has doubts that whether such data can well represent the complexity of the real world.
2. It is also interesting to extend the existing benchmark to generative tasks like captioning.
3. Also, since the benchmark contains data of 10 modalities, it will be interesting to see whether the training data in one domain can help the learning of the other domain. For example, it seems that the learning of language can help with audio recognition. However, whether the learning of tabular data can help with the learning of audio recognition seems to be unclear. It will provide more insight to analyze how different kinds of data cooperates with each other and when they will benefit the learning of each other.

**Additional Feedback:**

None.

**Documentation:**

Yes.

**Ethics:**

No.

**Relation To Prior Work:**

Yes, it is.

**Summary And Contributions:**

This paper proposes a new benchmark for multimodal learning. It collects data of 10 modalities for 20 prediction tasks in 6 research areas. A set of models are implemented and analyzed. The benchmark targets at evaluating multimodal models' abilites including generalization, efficiency, and robustness.

---

> ### Author Response · Authors · 2021-07-11
> **Thank you for your feedback, we have revised our paper accordingly**
>
> Thank you for your valuable feedback and insightful comments! We respond to some concerns below:
>
> [classification tasks] We are genuinely committed to building a community around these resources and continue improving it over time. We chose to focus on multimodal fusion by design for this first version of MultiBench to have a more coherent way to standardize and evaluate methods across datasets. We acknowledge the breath of multimodal learning and are looking forward to expanding it in other directions. We already identified 3 candidates for this expansion: multimodal grounding (e.g., image, video, and table-based question answering), cross-modal translation (e.g., image captioning and retrieval) and reinforcement learning (including language-vision navigation and robotics). Before finalizing the next topics, we plan to involve the broader community and gather feedback. Having this first benchmark released will enable these future discussions and expansions.
>
> Although this first version of the dataset focused on prediction and fusion (referred to as classification tasks by the reviewer), we want to point out that these tasks are not simple in nature. Solving these tasks accurately and robustly requires modeling a complex set of interactions between modalities. We greatly value the importance of having benchmarks as close as possible to real-world scenarios, hence we emphasized this issue by defining, for example, evaluation metrics that go beyond accuracy and include measures of complexity and robustness.
>
> [captioning] As previously mentioned, cross-modal translation (which include generative tasks such as captioning) is one of the expansion directions we are actively exploring. We have already included 2 datasets in captioning (and more generally for non-language outputs, retrieval): (1) Yummly-28K of paired videos and text descriptions of food recipes [1], and (2) Clotho dataset for audio-captioning [2] in our latest GitHub commits (https://github.com/pliang279/MultiBench/commits/main). We have also included a language-guided RL environment Read to Fight Monsters (RTFM) [3] in the recent commits too (see details on added text in page 10, under 'Other multimodal research problems').
>
> Finally, we have also begun processing some landmark retrieval datasets such as Flickr30k [4] and multimodal robotics datasets (reinforcement learning task in the Vision & Touch dataset [5]) for future inclusion to MultiBench and are planning to include more recent datasets as well. For this purpose, we plan an open call so that we can get feedback from the community (see Appendix I.4).
>
> [one domain helping another domain] MultiBench was effectively created with the goal of exploring such research questions. We thank you for the suggestion. We already explored this line of research on a few datasets (e.g., CMU-MOSI) in our prior work [6] and found that learning multimodal representations from language, video, and audio (on CMU-MOSI dataset) can help in language learning (on SST & IMDB datasets). We plan to include a section about this research question in the revised version of the paper, emphasizing the great potential of co-learning across modalities and domains.
>
> Our recent initial experiments on transfer learning found that pre-training on larger datasets in the same domain (i.e., CMU-MOSEI, UR-FUNNY) can help improve performance on smaller datasets when fine-tuned on a smaller dataset (i.e., CMU-MOSI). We indeed found performance on the smaller CMU-MOSI to improve from 75.2 to 75.8 using the same late fusion model with transfer learning from the larger UR-FUNNY and CMU-MOSEI datasets (see details on added text in page 10, under 'Multimodal transfer learning and co-learning').
>
> We believe that these potential future studies in co-learning and transfer learning are in fact strengths of MultiBench since it enables easy experimentation of such research questions.
>
> [1] Weiqing Min, Shuqiang Jiang, Jitao Sang, Huayang Wang, Xinda Liu, and Luis Herranz. 2017. Being a Super Cook: Joint Food Attributes and Multi-Modal Content Modeling for Recipe Retrieval and Exploration. IEEE Transactions on Multimedia 2017
>
> [2] Konstantinos Drossos, Samuel Lipping, and Tuomas Virtanen. Clotho: An audio captioning dataset. ICASSP 2020
>
> [3] Victor Zhong, Tim Rocktäschel, and Edward Grefenstette. RTFM: Generalising to new environment dynamics via reading. ICLR 2020
>
> [4] Peter Young, Alice Lai, Micah Hodosh, and Julia Hockenmaier. From image descriptions to visual denotations: New similarity metrics for semantic inference over event descriptions. TACL 2014.
>
> [5] Michelle A Lee, Yuke Zhu, Krishnan Srinivasan, Parth Shah, Silvio Savarese, Li Fei-Fei, Animesh Garg, and Jeannette Bohg. Making sense of vision and touch: Self-supervised learning of multimodal representations for contact-rich tasks. ICRA 2019.
>
> [6] Amir Zadeh, Paul Pu Liang, and Louis-Philippe Morency. Foundations of multimodal co-learning. Information Fusion, 64:188–193, 2020.

---

### Official Review · Reviewer_nPgz · 2021-07-05
**This work introduces a large-scale unified dataset involving distinct research areas, such effort is a good trial.**

**Rating:** 7
**Confidence:** 3
**Clarity:** Yes, it is easy to follow.

**Strengths:**

1.	This dataset integrates previous efforts towards multi-modal learning, covering diverse domains, and could serve as a good basis for benchmarking methods.
2.	There are novel ideas and concerns for the use of this dataset, including qualifying the time and space complexity, assessing the robustness when some modalities are missing, etc. All these could provide more information about the advantages/disadvantages of a given model from different aspects instead of simply considering quantitative results.
3.	Code for various classical multi-model methods is provided, ensuring reproducibility and facilitating future research. Standard evaluation tools are provided to ensure a fair comparison.
4.	Detailed analysis of different factors is conducted (e.g. generalization, performance, complexity, etc.). Limitations are discussed.


**Weaknesses:**

In each of the 6 distinct research areas, the annotation modality and focus of each chosen dataset may not be consistent. For example, AV-MNIST and Kinetics are both grouped into “multimedia”, but is it necessary for a model which could understand human actions well to be good at recognizing handwritten digits?



**Additional Feedback:**

When unifying datasets, semantic granularities are suggested to be considered.

**Correctness:**

Are there some logical rules for choosing these research areas? The spanning of these research areas should be systematical with some underlying hierarchies (such as ImageNet).

**Documentation:**

Code of various baselines is released. A standard public toolkit is provided. Details are sufficient. One thing is that a user must download each of the containing datasets by himself, this seems a little inconvenient. Though some datasets have restricted access, pre-extracted features could be provided.

**Ethics:**

There are no ethical concerns.

**Relation To Prior Work:**

This dataset involves multiple research areas, which makes it distinguished from datasets focusing on a single domain. The scale and the number of tasks it could support are also impressively large.

**Summary And Contributions:**

This paper unifies a large number (up to 15) of current multi-modal datasets involving 10 modalities and 6 research areas. Being able to support a variety of multi-modal tasks. A comprehensive toolkit called MULTIZOO is released, containing 20 different methods, serving as a good starter code.

---

> ### Author Response · Authors · 2021-07-11
> **Thank you for your feedback, we have revised our paper accordingly**
>
> Thank you for your valuable feedback and insightful comments! We respond to some concerns below:
>
> [multimedia] We welcome this feedback about how to taxonomize the different datasets. As pointed out by the reviewer, a significant topic in multimedia research is related to human action understanding. When using the term Multimedia, we used a somewhat broader definition, closer to the definition of ACM Multimedia conference. Historically, multimedia research included a strong computer vision component. We have revised our paper by clarifying our methodology for grouping datasets and we will also reassess our research grouping,
>
> [research areas] We chose these research areas through a survey of recent research papers in multimodal learning across conferences in machine learning and beyond (e.g., HCI, NLP, vision, robotics conferences). Furthermore, we consulted with domain experts in applying multimodal learning to their respective application areas to determine areas of large potential. Through engaging with domain experts we were able to select research areas and datasets that reflected realism in data collection, input modalities, preprocessing, and tasks which present challenges for machine learning models and potential for real-world transfer of learned algorithms.
>
> We plan to release an open call to the community for task and dataset proposals which will also help us expand into new research areas as proposed by domain experts. In the public GitHub (https://github.com/pliang279/MultiBench), we have included a section on contributing to MultiBench through either task proposals or additions of datasets and algorithms. The authors will regularly monitor new proposals through this channel. We have also added details regarding an open call to the community in Appendix I.4. Since MultiBench is publicly released and will be regularly maintained, we believe that the existing starting benchmark, code, evaluation, and experimental protocols can greatly accelerate the addition of new research areas in the future.
>
> [documentation] We have provided clear instructions to download all datasets in our public GitHub repository: https://github.com/pliang279/MultiBench. For almost all datasets, a single python command (get_data.py or download_data.sh) downloads the dataset or preprocessed features. The only exception is the MIMIC dataset in healthcare, where users are required to complete a training before they can access the dataset or features (which can be obtained via an email). Otherwise, we have provided all preprocessing code which makes loading these datasets very easy. We aim to ensure future datasets in MultiBench will also be simple to download (one line command) and include complete preprocessing code for direct machine learning experimentation.

---

### Author Response · Authors · 2021-07-11
**Thank you to all reviewers for your valuable feedback - summary of revisions made to submission**

Dear all reviewers, we are grateful for your valuable feedback and insightful comments. We are glad that you all found such a standardized multimodal benchmark, implementation of algorithms, and evaluation suite to be greatly useful for the community. We are genuinely committed to building a community around these resources and continue improving it over time. Your concrete suggestions are a valuable step in this direction, and we have revised our submission accordingly to take these into account. In this short note we summarize the main changes to the latest revision of our submission (the main additions are on page 10 and the final page of the Appendix section I.4, we have also included the updated Appendix to the back of the main paper submission so the full paper can be viewed from the pdf link). All updates are highlighted in red:

1. New datasets outside of fusion/classification (reviewer jKQc): We have already included 2 datasets in captioning/retrieval: (1) Yummly-28K of paired videos and text descriptions of food recipes [1], and (2) Clotho dataset for audio-captioning [2] and a language-guided RL environment Read to Fight Monsters (RTFM) [3] in the recent GitHub commits (https://github.com/pliang279/MultiBench/commits/main). We also describe some concrete new datasets we are currently integrating into MultiBench in page 10, under 'Other multimodal research problems'.

2. Open call to the community (reviewer xwdp): We agree that an open call to the community for task and dataset proposals is a scalable method in the expansion of MultiBench. In Appendix I.4, we added concrete plans to use MultiBench as a theme for future workshops and competitions and offer an open-call for suggestions and feedback about domains, datasets and metrics.

3. Transfer learning and co-learning (reviewer jKQc): Our recent experiments on transfer learning found that performance on the smaller CMU-MOSI improved from 75.2 to 75.8 using transfer learning from the larger UR-FUNNY and CMU-MOSEI datasets (see details on added text in page 10, under 'Multimodal transfer learning and co-learning').

4. Multitask learning (reviewer xwdp): We added results on page 10, under 'Multitask learning across modalities'. We found promising results in training a single model for all 4 datasets in the affective computing domain with performance on the largest CMU-MOSEI dataset improving from 82.1 to 82.9 using a multimodal transformer model. We believe that these potential future studies in transfer learning and multi-task learning are in fact strengths of MultiBench since it enables easy experimentation of such research questions.

5. Clarification of dataset and research area selection (reviewer nPgz): In Appendix C.1 'Dataset Selection' we have added more details in the selection of research areas and plans for future expansion to new research areas.

---

### Decision · Program_Chairs · 2021-07-26

**Decision:**

Accept

**Comment:**

This paper proposes a multi-scale benchmark integrating information from multiple heterogeneous sources of data. It is an important area and a very useful contribution to the community to assess the generalization capabilities of the ML models across domains/modalities and modality robustness (robustness to noisy and missing modalities.)

All the reviewers suggested the paper for acceptance without any significant concerns. It seems like the authors of the paper considered the reviewers' suggestions and improved the paper to include:
1) New datasets for retrieval and classification addressing the concerns pointed by Reviewer jKQc.
2) Transfer learning and co-learning results are addressing the comment made by Reviewer jKQc.
3) Multitask learning results addressing the feedback by Reviewer xwdp.
4) Clarifications addressing the concerns of Reviewer nPgz, and others.
5) Call for contributions from the community addressing the feedback by Reviewer xwdp.

I think the updates and the paper's responses in their current form address some of the important issues raised by the reviewers and I think this benchmark can be a good contribution to the multi-modal learning community.